# Multi-syndrome, multi-gene risk modeling for individuals with a family history of cancer with the novel R package PanelPRO

Gavin Lee[1†], Jane W Liang[2,3†], Qing Zhang[4], Theodore Huang[2,3], Christine Choirat[1], Giovanni Parmigiani[2,3], Danielle Braun[2,3]*

[1]Swiss Data Science Center, ETH Zürich and EPFL, Lausanne, Switzerland; [2]Department of Biostatistics, Harvard T.H. Chan School of Public Health, Boston, United States; [3]Department of Data Sciences, Dana-Farber Cancer Institute, Boston, United States; [4]Broad Institute of MIT and Harvard, Cambridge, United States

**Abstract** Identifying individuals who are at high risk of cancer due to inherited germline mutations is critical for effective implementation of personalized prevention strategies. Most existing models focus on a few specific syndromes; however, recent evidence from multi-gene panel testing shows that many syndromes are overlapping, motivating the development of models that incorporate family history on several cancers and predict mutations for a comprehensive panel of genes.

We present PanelPRO, a new, open-source R package providing a fast, flexible back-end for multi-gene, multi-cancer risk modeling with pedigree data. It includes a customizable database with default parameter values estimated from published studies and allows users to select any combinations of genes and cancers for their models, including well-established single syndrome BayesMendel models (BRCAPRO and MMRPRO). This leads to more accurate risk predictions and ultimately has a high impact on prevention strategies for cancer and clinical decision making. The package is available for download for research purposes at https://projects.iq.harvard.edu/bayesmendel/panelpro.

*For correspondence:
bmendel@jimmy.harvard.edu

†These authors contributed equally to this work

Competing interests: The authors declare that no competing interests exist.

## Introduction

In the last decade, DNA sequencing has changed dramatically. Tests have become faster and more affordable, leading to discovery of a growing number of germline pathogenic variants associated with increased cancer risk. Multi-gene panels are routinely available and include varying combinations of genes (*Plichta et al., 2016*). Evidence is accruing that gene mutations, which were typically believed to be only associated with one or two types of hereditary cancers, may in fact increase the risk for a wider range of syndromes. These advancements have changed the genetic counseling landscape by introducing a need to consider a wider set of individual genes and cancers to accurately assess overall risks. In the context of genetic counseling, the importance of accurate estimates of carrier probabilities is well-known (*Nelson et al., 2014*). With the number of genes of interest and their combinations increasing, efficient calculation of these estimates becomes crucial in clinical settings.

In genetic counseling, an individual may be suspected of inherited cancer susceptibility if their family history exhibits certain patterns. For example, if two or more relatives have the same type of cancer on the same side of the family, or if cancer diagnoses in the family are particularly early, they may be referred to testing for mutations in genes associated with increased risk for those specific

**eLife digest** Genetic mutations that increase cancer risk can be passed down from parents to their children, which can affect families across many generations. In these families, multiple members may be affected by different types of cancer, and these cancers often develop at an early age. Unaffected family members are often referred to genetic counselling, where they can explore their own risk of cancer. Clinicians and genetic counselors can provide recommendations to minimize cancer risk and inform personal choices on how to manage that risk, such as opting for preventative surgeries or participating in regular screening.

In genetic counselling sessions, highly trained clinicians and specialists use software that takes an individual's family history of cancer and uses it to estimate their individual risk of carrying certain genetic mutations. These estimates can in turn help to predict their future risk of cancer. Many existing software packages are limited to estimating risks based on mutations in well-known cancer-related genes, such as *BRCA1* and *BRCA2* in breast and ovarian cancer. However, emerging evidence suggests that many of the genes associated with cancer risk work as part of a complex and overlapping network. Since current risk-profiling software packages are only designed to consider such genes in isolation, they cannot generate the most robust, accurate or comprehensive cancer risk profiles.

To address this challenge, Lee, Liang et al. have developed a new risk-profiling software that can integrate a large number of gene mutations and a wide range of potential cancer types to provide more accurate estimates of individual cancer risk. This software, called PanelPRO, uses evidence identified from extensive literature reviews to model the complex interplay between genes and cancer risk. The software not only calculates risks based on known genes, but also allows other developers to integrate new cancer-related genes that may be identified in the future. Importantly, the software is compatible with genetic counselling applications, since it returns answers within seconds when reasonable family and gene database sizes are used.

PanelPRO is a new, modern, flexible and efficient software package that provides an important advance towards modelling the vast genetic and biological complexity that contributes to inherited cancer risk. This software is designed to provide a more accurate and comprehensive estimate of cancer risk for individuals with family histories of cancer.

As an open-source software, it is freely available for research purposes, and can be licensed by software companies and healthcare organizations to integrate electronic patient records and rapidly identify at-risk individuals across larger patient groups. Ultimately, this software has the potential to improve cancer prevention strategies and optimize the personalized decision-making processes around cancer risk.

cancers. In the case of hereditary breast cancer, guidelines in the US were established to identify patients who have higher likelihoods of benefiting from germline genetic testing. Thresholds for testing were set high initially, since genetic testing was very expensive at the time (*Manahan et al., 2019*). Although cost of testing has decreased and guidelines are constantly changing, accurate calculation of carrier probabilities, given family history, is essential in supporting the decision for further testing, preventative treatment, or family planning (*Chen et al., 2004*).

Existing Mendelian models consider a relatively narrow subset of cancers and genes. For example, BRCAPRO, available in the BayesMendel R package (*Chen et al., 2004*), considers two cancers (breast and ovarian) and two genes (BRCA1 and BRCA2). Boadicea v4 Beta (*Centre for Cancer Genetic Epidemiology, 2020*) considers BRCA1, BRCA2, PALB2, CHEK2, and ATM mutations in the same cancers. To comprehensively incorporate cancers, genes, and their interactions, we introduce PanelPRO, an R package which aims to efficiently and flexibly scale to the demands of germline panel testing. The newly developed package has the following key advantages:

- Customizable model specification, including the choice of genes and cancers included in the model;
- Customizable model parameters, including the allele frequencies and penetrances;
- Accurate default parameter estimates, curated from an extensive literature search;
- Flexibility to incorporate cancer risk modifiers such as prophylactic surgeries;

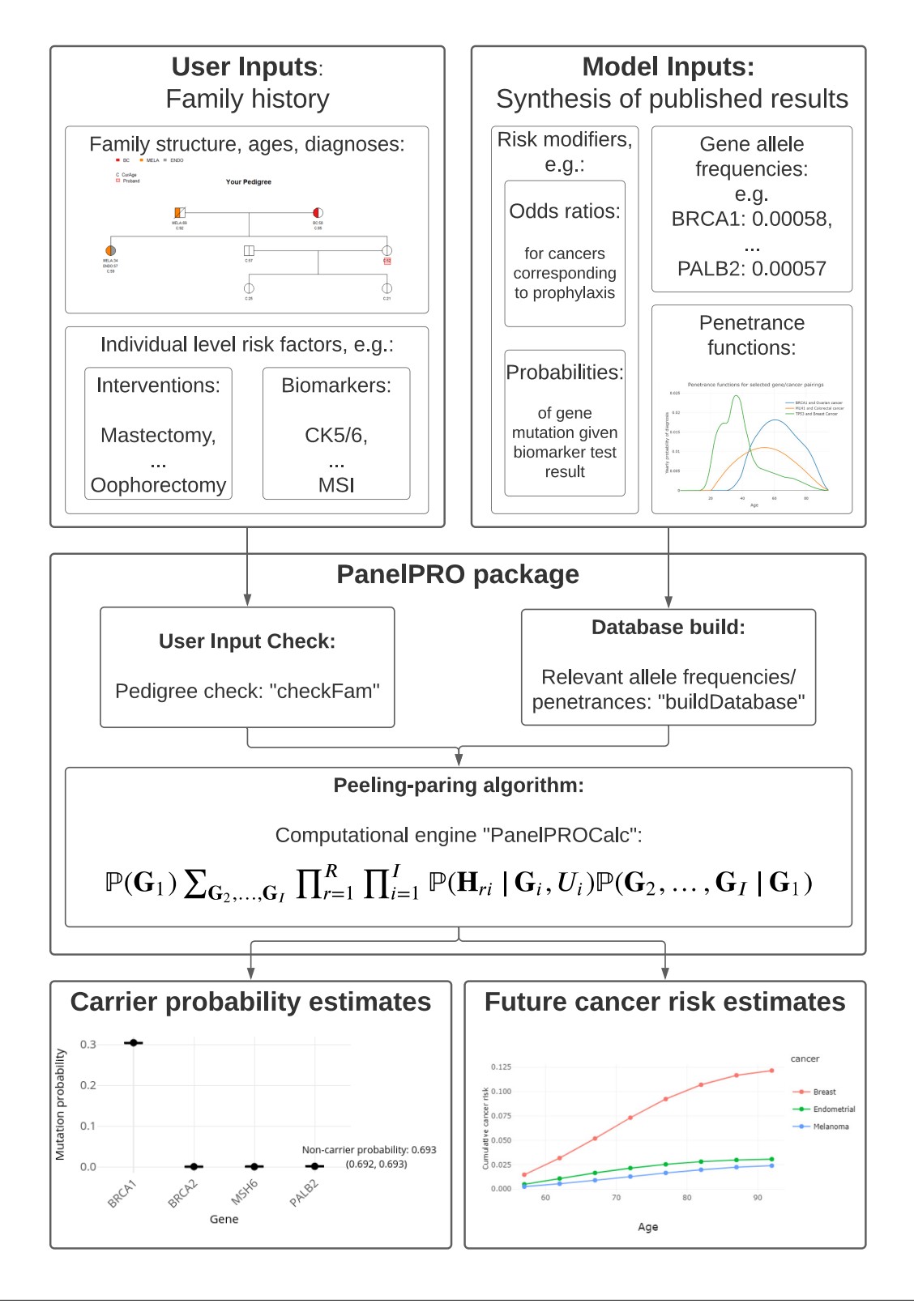

**Figure 1.** PanelPRO package workflow.

- Comprehensive user input checks for pedigrees;
- Speed of computation, through an optimized C++ implementation of an efficient algorithm. (*Madsen et al., 2018*)

In general, PanelPRO can handle models with $K$ genes and $R$ cancers, where $K$ and $R$ are arbitrary (subject to reasonable run-times and memory constraints). It is intended that $K$ and $R$ become larger as more research on gene cancer associations becomes available. The package contains a comprehensive collection of functions designed to efficiently calculate carrier probabilities and future cancer risk for individuals, given detailed information about their family history.

PanelPRO is designed to be back-compatible with the existing BayesMendel package; individual models within that package (for example, *BRCAPRO, MMRPRO, BRCAPRO5*, or *BRCAPRO6*) can be called directly from PanelPRO by passing this model specification to the main function call. We expect that users of BayesMendel will migrate to this generalized and customizable enhancement, and that PanelPRO will lead to new users interested in broader cross syndrome modeling in the current landscape for cancer clinical risk assessment. In the current release, there are minor differences in how BayesMendel and PanelPRO deal with peer-reviewed data, in particular, for cancer penetrance calculations.

The PanelPRO R package is freely available for research purposes. The BayesMendel R package is likewise freely available for research purposes, and is currently licensed for clinical commercial use to CRA Health, CancerGene Connect, Progeny, FamHis, MagView, Igentify, CancerIQ, and Finch genetics. Family history has long been understood to be a key component for identifying risk and preventing heritable diseases, and clinical tools such as the ones which license BayesMendel are becoming more readily available (*Welch et al., 2018*). For BayesMendel, we leave clinical integration to the software licensees, including the integration of electronic medical records such as EPIC. We envision a similar dissemination plan for PanelPRO.

**Table 1.** Pedigree structure in PanelPRO.

| Column | Definition | Value |
|---|---|---|
| ID | Unique numeric identifier of each individual | Non-repeated strictly positive integer |
| MotherID | ID of one's mother | Strictly positive integer or NA (missing) |
| FatherID | ID of one's father | Strictly positive integer or NA (missing) |
| Sex | Sex of the individual: 1 for male, 0 for female | One of {0, 1} |
| isProband | Indicates the proband or counselee by 1 and 0 otherwise – multiple probands can be specified | One of {0, 1) |
| CurAge | Age of censoring: either the current age or death age, depending on isDead status | Positive integer or NA (missing) |
| isAff* | Affection status of cancer * | One of {0, 1} |
| Age* | Affection age of cancer * | Positive integer or NA (missing) |
| isDead | Whether someone has died | One of {0, 1, NA} |
| race | Race of individual (used to modify penetrance) | One of All_Races, AIAN, Asian, Black, White, Hispanic, WH, WNH, NA |
| Ancestry | Ancestry of individual (used to modify allele frequencies) | One of AJ, nonAJ, Italian, NA |
| Twins | Identifies siblings who are identical twins or multiple births | Each set is identified by a unique integer, and 0 otherwise |
| riskmod | Preventative interventions which modify penetrance | List, combination of "mastectomy", "hysterectomy", and "oophorectomy" |
| InterAge | Age of each preventative interventions | List, combination of integers |
| Gene name from GENE_TYPES | Germline testing result | One of {0, 1, NA} |
| Marker name from CK14, CK5.6, ER, PR, HER2, MSI | Marker testing result | One of {0, 1, NA} |

**Table 2.** List of model options that the user can pass to PanelPRO, along with their defaults.

| Option | Default value | Possible values | Description |
|---|---|---|---|
| max.mut | NULL | Integers up to the number of genes | Number of maximum simultaneous mutations, also known as the paring parameter. If no integer has been input, it re-defaults to 2. |
| iterations | 20 | Integers from 1 upwards | In case of missing current or cancer ages in the pedigree, this is the number of times those ages will be imputed. |
| parallel | TRUE | TRUE or FALSE | If age imputations are needed, this parameter can be set to utilize multiple cores on one's machine. |
| net | FALSE | TRUE or FALSE | Determines whether net or crude penetrances are used to compute future risk of cancer. Net penetrances exclude all other causes of death, apart from the affected cancer. |
| age.by | 5 | Integers from one upwards | The intervals of age used to report the future risk of cancer. |

## Methods: package workflow

The workflow of the package includes four main parts: the input, including user and model input; pre-processing of the inputs, including user input checks and a database build; running the peeling-paring algorithm; and outputting the results. *Figure 1* shows the workflow. Additionally, *Appendix 1—figure 1* shows detailed sub-routines within the package.

## Input

### User input

The main input from the user is their pedigree. This is in the form of an R `data.frame` which contains detailed information about known family members, such as their ages and previous cancer diagnoses. The pedigree structure is defined by the `ID`, `MotherID` and `FatherID` columns. Previous cancer diagnoses and their ages of diagnosis are stored in the `isAff*` and `Age*` columns, respectively, where * represents a cancer type, designated according to a standard nomenclature of two- to four-letter tags which are also used in visualizations. Cancers currently considered in Panel-PRO are listed in *Appendix 1—table 1*. Note, this list will be expanded for future versions of the package as more information on risk for various genes and cancers becomes available in the literature. Risk modifiers, such as prophylactic surgeries, can be incorporated to adjust the likelihood

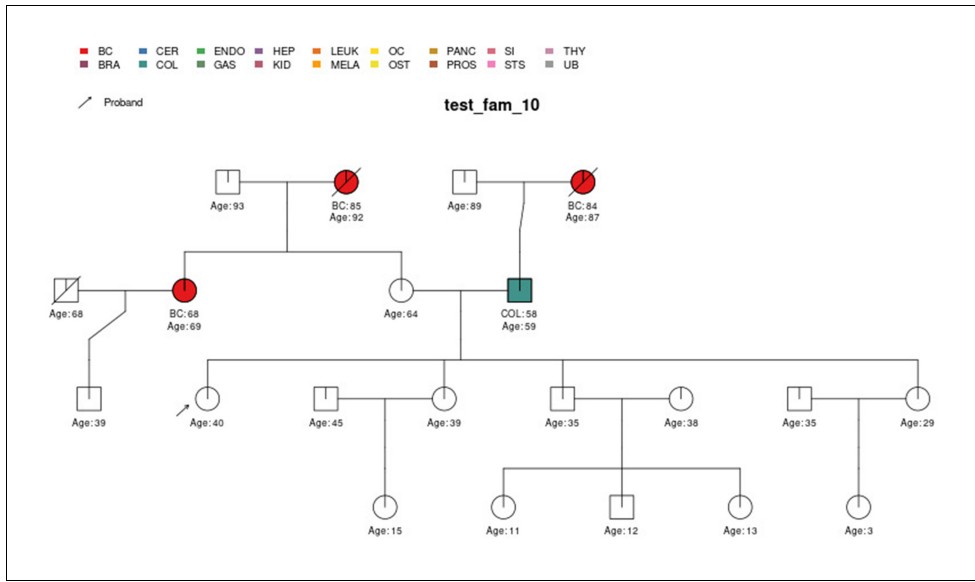

**Figure 2.** *test_fam_1* sample pedigree as included in the PanelPRO package, plotted using the external visPed package. The colors refer to cancer diagnoses in the legend. The age of diagnosis is shown below the individual if it is known.

calculation. Previous genetic testing history can also be incorporated. The current version of the package, v0.2.0, contains thirteen sample pedigrees, called *test_fam_X*, where *X* goes from 1 to 12, and *err_fam_1*. The *test_fam_X* pedigrees provide realistic and extreme examples of the data that can be included in the user input, whereas *err_fam_1* is an example of a family that does not pass PanelPRO's preprocessing pedigree check and therefore cannot be evaluated by the model. A clipped version (with a subset of the necessary columns) of *test_fam_1* pedigree is shown below. Key figures relevant to the pedigree can be found in *Appendix 1—table 2*.

**head**(test_fam_1)

| ## | | ID | Sex | MotherID | FatherID | isProband | CurAge | isAffBC | isAffBC | AgeBC | AgeOC | isDead |
|----|----|----|-----|----------|----------|-----------|--------|---------|---------|-------|-------|--------|
| ## | 1 | 1 | 0 | NA | NA | 0 | 93 | 1 | 0 | 65 | NA | 1 |
| ## | 2 | 2 | 1 | NA | NA | 0 | 80 | 0 | 0 | NA | NA | 1 |
| ## | 3 | 3 | 0 | 1 | 2 | 0 | 72 | 1 | 1 | 40 | NA | 0 |
| ## | 4 | 4 | 1 | 1 | 2 | 0 | 65 | 0 | 0 | NA | NA | 1 |
| ## | 5 | 5 | 1 | 1 | 2 | 0 | 65 | 0 | 0 | NA | NA | 0 |
| ## | 6 | 6 | 0 | 1 | 2 | 1 | 55 | 0 | 0 | NA | NA | 0 |

The full specification of the pedigree structure is shown in *Table 1*. The family tree pedigree can also be visualized by using the visPed package as in *Figure 2*. This external package is available through https://github.com/bayesmendel/visPed version 0.1.0 (*Lee, 2021*) and is based on the kinship2 package available in CRAN (*Sinnwell et al., 2014*). PanelPRO itself does not contain pedigree plotting functionality. However, users can easily acquire the visPed package separately.

Prophylactic surgeries (mastectomy, oophorectomy and hysterectomy) act as risk modifiers. They are specified in a column of lists in the user input pedigree. Including these risk modifiers changes the resulting carrier probability and future risk outputs (see the Output section). Previous history of biomarker testing for breast and colorectal cancers can also be included in the model.

## Model input

Calculation of carrier probabilities requires information about allele frequencies and penetrances for the mutations and cancers requested in the function call. They are derived from peer-reviewed studies whose results are cataloged in the PanelPRODatabase. At each function call, the code extracts the appropriate subset of gene-cancer combinations. These combinations are specified in the main PanelPRO function call, where the user should indicate the cancers for which family history in the pedigree should be used, as well as the genes for which carrier probabilities are requested.

```
PanelPRO(pedigree = test_fam_1,
         cancers = c('Breast', 'Ovarian'),
         genes = c('BRCA1', 'BRCA2', 'ATM', 'MSH2'))
```

If no genes or cancers are specified, `PanelPRO` will default to all the supported genes in the version at that time, with all the cancers in the pedigree.

Users are free to change the database defaults for their own purposes. The structure of this database is an R list. A partial output is provided below.

**str(PanelPRODatabase)**

```
##       $ Penetrance          : num [1:18, 1:26, 1:8, 1:2, 1:94, 1:2] 3.98e-05 2.80e-07 0.00 5.00e-08 0.00 ...
##        ..- attr(*, 'dimnames')=List of 6
##        .. ..$ Cancer          : chr [1:18] 'Brain' 'Breast' 'Cervical' 'Colorectal' ...
##        .. ..$ Gene            : chr [1:26] 'APC_hetero_anyPV' 'ATM_hetero_anyPV' 'BARD1_hetero_anyPV' ...
##        .. ..$ Race            : chr [1:8] 'All_Races' 'AIAN' 'Asian' 'Black' ...
##        .. ..$ Sex             : chr [1:2] 'Female' 'Male
```

*Continued on next page*

```
##      .. ..$ PenetType        : chr [1:2] 'Net' 'Crude
##         $ AlleleFrequency    : num [1:24, 1:3] 1.45e-04 1.90e-03 3.41e-04 2.17e-05 1.37e-02 ...
##      ..- attr(*, 'dimnames')=List of 2
##      .. ..$ Gene             : chr [1:24] 'APC_anyPV' 'ATM_anyPV' 'BARD1_anyPV' 'BMPR1A_anyPV' ...
##      .. ..$ Ancestry         : chr [1:3] 'AJ' 'nonAJ' 'Italian'
```

In the current PanelPRO database, cancer penetrances are taken from data included in the Bayes-Mendel package when available: the BRCA1 and BRCA2 estimates for the probability of developing breast or ovarian cancer (*Chen et al., 2020*); the MLH1, MSH2, and MSH6 estimates for the probability of developing colorectal or endometrial cancer (*Wang et al., 2020*; *Felton et al., 2007*); and the CDKN2A estimates for the probability of developing melanoma (*Wang et al., 2010*; *Begg et al., 2005*; *Bishop, 2002*). All other cancer penetrances are pulled from the All Syndromes Known to Man Evaluator (ASK2ME) clinical tool (*Braun et al., 2018*).

For allele frequencies, we use the non-Ashkenazi, Ashkenazi Jewish, and Italian BRCA1 and BRCA2 allele frequency estimates from BRCAPRO (*Chen et al., 2004*; *Antoniou et al., 2002*); for MLH1, MSH2, and MSH6, we use the allele frequency estimates from MMRpro (*Chen et al., 2004*; *Chen et al., 2006*); and for CDKN2A, we use the allele frequency estimate from Melapro (*Chen et al., 2004*; *Berwick, 2006*). Allele frequency estimates for ATM, CHEK2, and PALB2 are taken from *Lee et al., 2016*. The allele frequencies of the remaining genes are estimated based on a 25-gene panel study of 252,223 individuals (*Rosenthal et al., 2017*) that did not adjust for ascertainment. In this case, we rescale the reported estimates by the ratio of the ascertained and unascertained allele frequencies for a gene reported in both our existing database and the study.

New genes and cancers will be added to PanelPRO based on regular literature reviews as conducted in ASK2ME (*Braun et al., 2018*). The ASK2ME approach identifies best-available studies that adjust for ascertainment; since many papers report odds ratios or relative risks, it then calculates absolute age-specific cancer penetrances when necessary.

The user can also select other options in the function call which are relevant at run-time. Examples include the maximum number of simultaneous gene mutations considered for a given individual, whether a parallelized version of the algorithm is performed, and the number of imputations in case of missing age data (see the Missing Data section). Many of the useful options are listed in *Table 2*.

Passing these options to the function call is simple, as shown below.

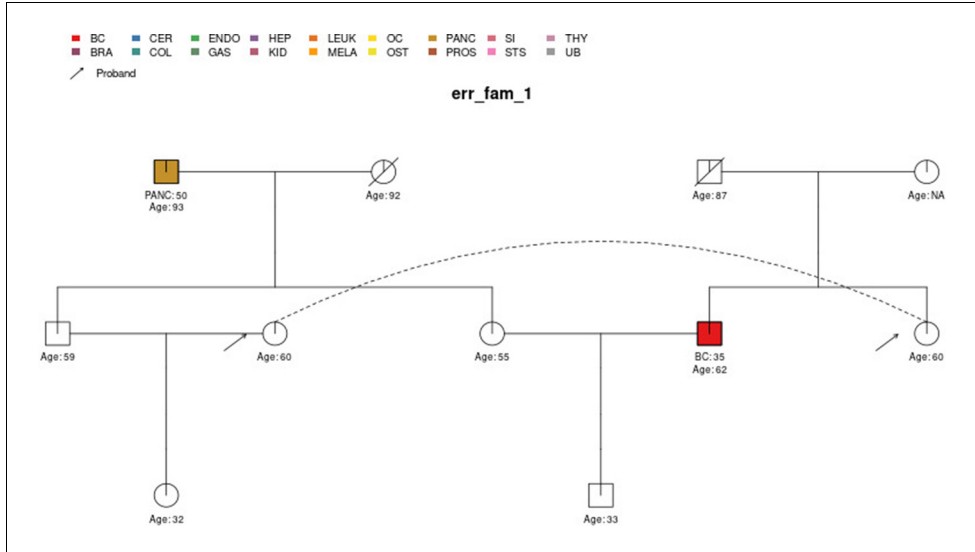

**Figure 3.** The sample pedigree err_fam_1 which contains a pedigree loop, due to the mating pattern of the siblings aged 59 and 55 with the siblings aged 60 and 62, respectively. The two circles linked by a dotted line represent the same individual.

```
PanelPRO(pedigree = test_fam_1,
         cancers = c('Breast', 'Ovarian'),
         genes = c('BRCA1', 'BRCA2', 'ATM', 'MSH2'),
         max.mut = 1,
         parallel = FALSE)
```

Instead of specifying a set of cancers and genes, users can call models corresponding to those in the BayesMendel package, as well as other predefined models. For example, the two calls below are equivalent.

```
bayesMendelCall <- BRCAPRO6(pedigree = test_fam_1)
panelProCall <- PanelPRO(pedigree = test_fam_1,
                    cancers = c('Breast', 'Ovarian'),
                    genes = c('BRCA1', 'BRCA2', 'MLH1', 'MSH2', 'MSH6',
'CDKN2A'))
all.equal(bayesMendelCall, panelProCall)
## (1) TRUE
```

## Preprocessing
### Pedigree check
First, PanelPRO checks the structure of the user-supplied R *data.frame* containing the family history to be evaluated, using a call to the checkFam function. A description can be found in *Table 3*. Messages or warnings are given to the user if values have been automatically changed to rectify conflicts. For example, *test_fam_1* contains some ancestry and race inconsistencies.

```
checkFam(test_fam_1)
## Your model has two cancers - Breast, Ovarian and 24 genes - APC_hetero_anyPV,
ATM_hetero_anyPV ...
## Germline testing results for BRCA1 are assumed to be for default variant BRCA1_-
hetero_anyPV.
## Germline testing results for BRCA2 are assumed to be for default variant BRCA2_-
hetero_anyPV.
## ID 3 's Ancestry has been changed to nonAJ to meet heredity consistency
## ID 10 's race has been changed to All_Races to meet heredity consistency
## ID 9 's Ancestry has been changed to nonAJ to meet heredity consistency
## ID 16,17 's Ancestry has been changed to nonAJ to meet heredity consistency
## ID 29,30 's Ancestry has been changed to nonAJ to meet heredity consistency
## ID 33,34 's race has been changed to All_Races to meet heredity consistency
## ID 33,34 's Ancestry has been changed to nonAJ to meet heredity consistency
```

Errors are given if inconsistencies or ambiguities in the pedigree cannot be resolved such that the pedigree can be safely passed into downstream functions. Most of the checks are for the presence of required information; whether the pedigree variables have values in the expected range/set; and for consistency between cancers and sex and in terms of features with hereditary assumptions among parents/children and twins. The pedigree is also checked for 'loops', which PanelPRO currently does not support. For example, within *err_fam_1*, there are two sets of male and female siblings (four individuals) who have mated with the corresponding siblings in another family, as shown in *Figure 3*. This mating configuration results in a loop. For a more detailed definition of loops, see *Fernando et al., 1993*. In addition, disconnected family members are detected and removed from the pedigree if they will not influence the counselee's results.

## Build database

Depending on the configuration of the model requested (cancers in the family, genes considered), a subset of `PanelPRODatabase` or a user-modified database will be created and passed through for further calculation by the buildDatabase function. A description of this function can be found in *Table 3* .

## Algorithm

The checked pedigree and PanelPRODatabase subset, as well as any user options, are then passed to the 'peeling-paring' algorithm, which approximates Equation 2 in the Genotype probabilities section. It is based on the 'peeling' algorithm as introduced by *Elston and Stewart, 1971* with its implementation based on *Fernando et al., 1993*. The 'paring' aspect of the algorithm limits the number of simultaneous mutations allowed. This is called the paring parameter and has a default value of 2, which results in an approximation which has been shown to be adequate for clinical purposes (*Madsen et al., 2018*). When the paring parameter is set equal to the number of distinct genes to be considered, the calculation is exact (assuming no other missing information about the pedigree). Future cancer risks are then calculated based on the law of total probability, using the previously calculated posterior carrier probabilities, as described in the Methods: Mendelian modeling section. These two calculations are performed in PanelPROCalc, as listed in *Table 3*. The underlying algorithm is written in *Rcpp* using the *RcppArmadillo* package (*Eddelbuettel and Sanderson, 2014*; *Eddelbuettel and Francois, 2011*). It uses, as much as possible, optimized data structures, vectorized operations and in-place modifications to be both time and memory efficient. See the Discussion section for benchmarks on the run-time of the implementation.

The recursive nature of the peeling-paring algorithm allows for multiple counselees to be specified in the function call without significant increase in the computational time. This is an advantage when multiple family members are at high risk and would benefit from knowing their carrier probabilities and future cancer risks.

## Missing data

PanelPRO calculates mutation carrier probabilities for one or more counselees. The peeling-paring algorithm requires both parents of the counselee to be present in the pedigree in order to link individuals who are non-founders. When there is only data for a single parent (whose children influence the results), we add a pseudo-parent who has the same prior allele frequencies as the parent for whom we do have information on. This allows the peeling-paring algorithm to run and serves as an approximation of the final results.

When the current age or age of cancer diagnosis of a family member is unknown, we use a multiple imputation procedure to repeatedly sample their age (*Biswas et al., 2013*). Unknown current ages are sampled based on the current ages of the relatives, and unknown ages of cancer diagnosis are sampled from the cancer penetrances, using the current age as an upper bound. The optional impute.times argument in the main PanelPRO function can be used to set the number of samples taken. The value labeled 'estimate' in the output is the average of the results over the sampled ages, whilst the 'lower' and 'upper' bounds are the minimum and maximum values over the respective samples (whether it be for the posterior probabilities or future risks).

When impute.times is high (say, 50 or more), it is recommended to set the parameter parallel to TRUE. The algorithm will then use the foreach package and the existing cores in one's machine to execute the imputations in a parallel fashion, instead of sequentially, thereby speeding up the computation.

## Output

For each proband in the pedigree, the output consists of:

- estimates of carrier probabilities,
- lower and upper bound estimates of carrier probabilities if imputations were made for missing data,
- estimates of future risks of cancers in 5-year intervals (the user can also change the length of the intervals),

- lower and upper bound estimates of future risks of cancers in 5-year intervals if imputations were made for missing data.

Messages or warnings generated from checkFam have been omitted in the example below for brevity.

```
output <- PanelPRO(pedigree = test_fam_1,
                   cancers = c('Breast', 'Ovarian'),
                   genes = c('BRCA1', 'BRCA2', 'ATM', 'MSH2'),
                   max.mut = 2,
                   parallel = FALSE)
## Your model has two cancers - Breast, Ovarian and four genes - BRCA1_hetero_anyPV
...
```

```
output
## $posterior.prob
## $posterior.prob$`6`
##                                               genes     estimate         lower         upper
## 1                                         noncarrier  6.895857e-01  6.852634e-01  6.901750e-01
## 2                                  BRCA1_hetero_anyPV  3.028050e-01  3.009048e-01  3.030639e-01
## 3                                  BRCA2_hetero_anyPV  3.486877e-04  2.817221e-04  4.216045e-04
## 4                                   ATM_hetero_anyPV  4.027689e-03  3.904178e-03  5.532214e-03
## 5                                  MSH2_hetero_anyPV  9.814770e-04  5.697486e-04  4.574016e-03
## 6          BRCA1_hetero_anyPV.BRCA2_hetero_anyPV  1.462717e-04  1.181799e-04  1.768601e-04
## 7           BRCA1_hetero_anyPV.ATM_hetero_anyPV  1.689866e-03  1.638046e-03  2.321095e-03
## 8            BRCA2_hetero_anyPV.ATM_hetero_anyPV  7.125399e-07  6.081245e-07  9.481190e-07
## 9           BRCA1_hetero_anyPV.MSH2_hetero_anyPV  4.118459e-04  2.390780e-04  9.481190e-07
## 10          BRCA2_hetero_anyPV.MSH2_hetero_anyPV  4.118459e-04  1.361573e-07  9.481190e-07
## 11            ATM_hetero_anyPV.MSH2_hetero_anyPV  2.541503e-06  1.637872e-06  9.481190e-07
##
##
## $future.risk
## $future.risk$`6`
## $future.risk$`6`$Breast
##
##     ByAge     estimate         lower         upper
## 1      60   0.04694233   0.04691714   0.04707834
## 2      65   0.09340834   0.09336108   0.09367413
## 3      70   0.13709254   0.13702686   0.13747579
## 4      75   0.17464537   0.17456501   0.17513057
## 5      80   0.20483919   0.20474732   0.20541009
## 6      85   0.22687210   0.22677195   0.22750817
## 7      90   0.23927170   0.23916714   0.23994509
##
## $future.risk$`6`$Ovarian
##     ByAge     estimate         lower         upper
## 1      60   0.03600079   0.03598267   0.17954734
```

*Continued on next page*

```
## 2                          65   0.07186345    0.07183188   0.17954734
## 3                          70   0.10464159    0.10460004   0.10477485
## 4                          75   0.13238354    0.13233432   0.13253217
## 5                          80   0.15520230    0.13233432   0.15536227
## 6                          85   0.17183044    0.17177152   0.15536227
## 7                          90   0.17960786    0.17954734   0.15536227
```

The package includes the function visRisk to visualize the output graphically. *Figure 4* demonstrates this usage for `test_fam_1`. The visRisk function was implemented using the plotly (*Sievert, 2020*) package, so that the output can be rendered interactively and display the exact probabilities upon hovering.

## Additional examples

In this section, we display some of the other test pedigrees included in PanelPRO and their corresponding output from the visRisk function, as well as a comparison to some other platforms and models. We compared PanelPRO to: BRCAPRO and MMRPRO in BayesMendel (*Chen et al., 2004*), CanRisk (*Lee et al., 2019*; *Carver et al., 2021*), IBIS (*Tyrer et al., 2004*), and PREMM-5 (*Kastrinos et al., 2017*), which all support different cancers, genes and model assumptions. *Table 4* summarizes the supported inputs and outputs of each model. If a particular model or platform does not support certain cancers, the irrelevant family history is simply omitted from input. For brevity, *Figures 5–9* provide visualizations of the pedigrees of these additional examples, and the corresponding model outputs are reported in as figure supplements. In all cases, we used the default model settings (if any). Many of the aforementioned platforms have long reports as outputs, so we have only included the portions concerned with carrier probabilities and future risks. The same information contained in the PanelPRO sample pedigrees is input to the other models; however not all of the features are used by these other platforms. Conversely, there are some inputs for the other models that PanelPRO does not include.

Notably, all the cancers and genes supported by these other models are a subset of those supported in PanelPRO, except for PREMM-5, which takes into consideration other cancers associated with Lynch syndrome which are not currently included in PanelPRO (bile duct and sebaceous gland).

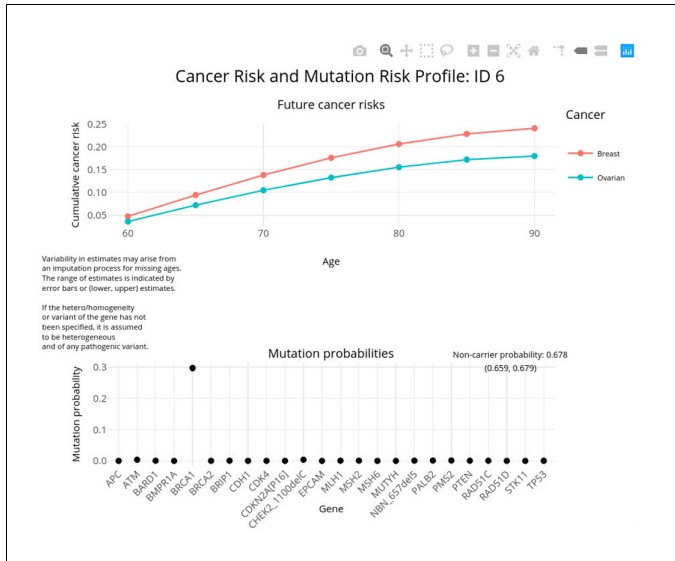

**Figure 4.** Sample output using visRisk function.

**Table 4.** Comparison between supported cancers and genes in PanelPRO and other platforms.

| Model or platform name | Version | Supported cancer input types | Supported gene carrier probability outputs | Supported future cancer risk outputs |
|---|---|---|---|---|
| PanelPRO | 0.2.0 | Brain, breast, cervical, colorectal, endometrial, gastric, kidney, leukemia, melanoma, ovarian, osteosarcoma, pancreatic, small intestine, soft tissue sarcoma, thyroid, urinary bladder, hepatobiliary | APC, ATM, BARD1, BMPR1A, BRCA1, BRCA2, BRIP1, CDH1, CDK4, CDKN2A, CHEK2, EPCAM, MLH1, MSH2, MSH6, MUTYH, NBN, PALB2, PMS2, PTEN, RAD51C, RAD51D, STK11, TP53 | same as cancer inputs |
| BRCAPRO | 2.1–7 | Breast, ovarian | BRCA1, BRCA2 | same as cancer inputs |
| MMRPRO | 2.1–7 | Colorectal, endometrial | MLH1, MSH2, MSH6 | same as cancer inputs |
| IBIS | 0.8b | Breast | NA | Breast |
| CanRisk | 1.2.3 | Breast, contralateral breast, ovarian, prostate, pancreatic | BRCA1, BRCA2, PALB2, CHEK2, ATM, RAD51D, RAD51C, BRIP1 | Breast, ovarian |
| PREMM-5 | NA | Colorectal, endometrial, other (group of ovarian, stomach, small intestine, urinary tract/bladder/kidney, bile ducts, brain, pancreas, sebaceous gland skin) | MLH1, MSH2, MSH6, PMS2, EPCAM | NA |

However, PREMM-5 does not provide the associated future risk for these additional cancers, only carrier probabilities for MLH1, MSH2, MSH6, PMS2, and EPCAM.

Comparing PanelPRO with BRCAPRO and MMRPRO, we see that PanelPRO offers carrier probability estimates for a larger set of genes, as well as a graphical output of the future risk. IBIS does not give any estimates for carrier probabilities; however, it gives a summary of the future risks in text format, relative to population averages. Finally, PREMM-5 gives an estimate of carrying any of five genes (MLH1, MSH2, MSH6, PMS2, or EPCAM), whilst PanelPRO is able to give estimates for each of those individual genes. PREMM-5 also does not give estimates for future risks of cancer.

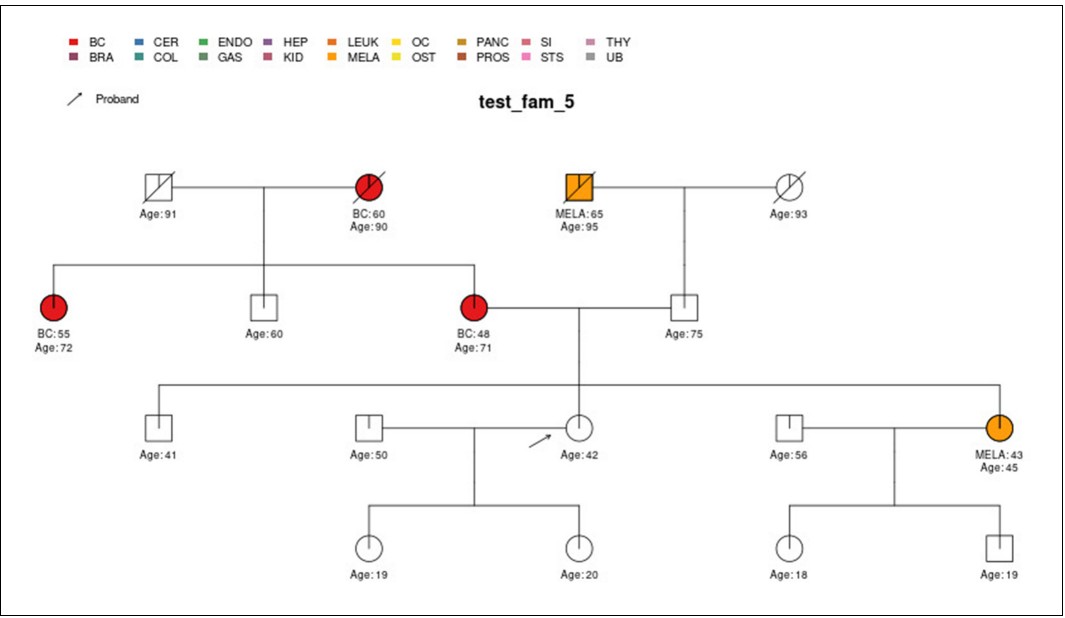

**Figure 5.** Sample pedigree *test_fam_5*.
The online version of this article includes the following figure supplement(s) for figure 5:

**Figure supplement 1.** PanelPRO output with *test_fam_5* as pedigree input.
**Figure supplement 2.** BRCAPRO output with *test_fam_5* as pedigree input.
**Figure supplement 3.** IBIS output with *test_fam_5* as pedigree input.

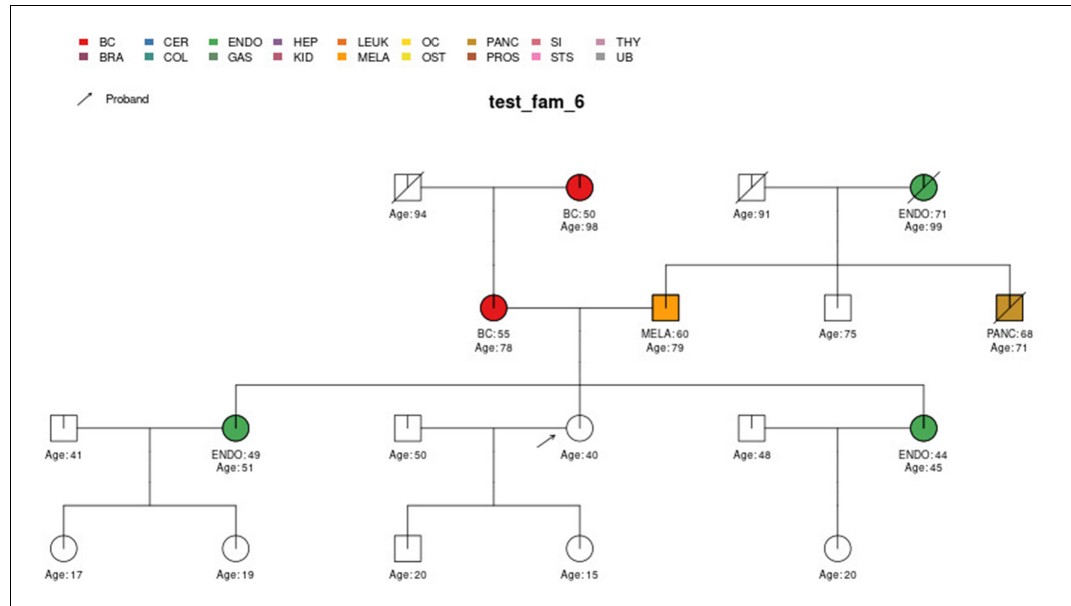

**Figure 6.** Sample pedigree *test_fam_6*.

The online version of this article includes the following figure supplement(s) for figure 6:

**Figure supplement 1.** PanelPRO output with *test_fam_6* as pedigree input.

**Figure supplement 2.** BRCAPRO output with *test_fam_6* as pedigree input.

**Figure supplement 3.** MMRPRO output with *test_fam_6* as pedigree input.

*Figure 6 continued on next page*

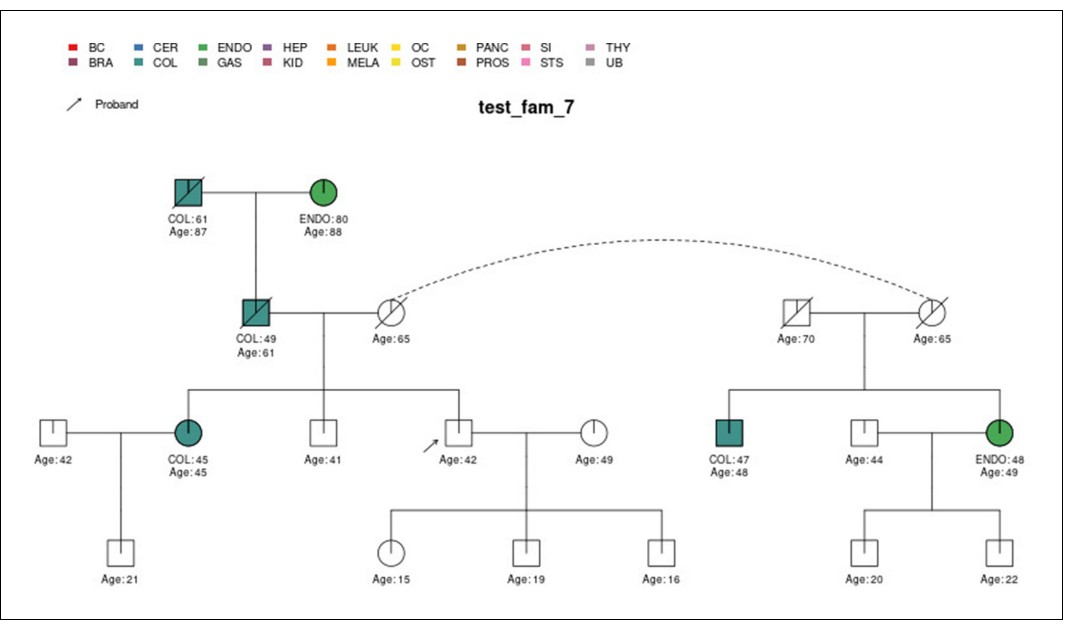

**Figure 7.** Sample pedigree *test_fam_7*.

The online version of this article includes the following figure supplement(s) for figure 7:

**Figure supplement 1.** PanelPRO output with *test_fam_7* as pedigree input.

**Figure supplement 2.** MMRPRO output with *test_fam_7* as pedigree input.

**Figure supplement 3.** PREMM-5 output with *test_fam_7* as pedigree input.

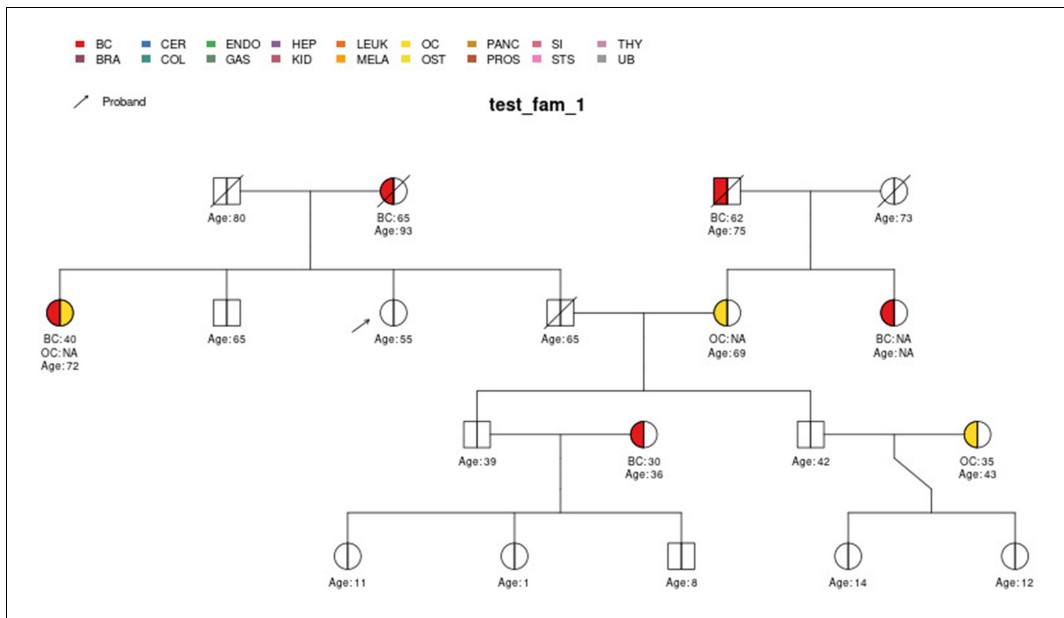

**Figure 8.** Sample pedigree *test_fam_10*.
The online version of this article includes the following figure supplement(s) for figure 8:

**Figure supplement 1.** PanelPRO output with *test_fam_10* as pedigree input.
**Figure supplement 2.** BRCAPRO output with *test_fam_10* as pedigree input.
**Figure supplement 3.** MMRPRO output with *test_fam_10* as pedigree input.
**Figure supplement 4.** IBIS output with *test_fam_1* as pedigree input.
**Figure supplement 5.** PREMM-5 output with test_fam_10 as pedigree input.

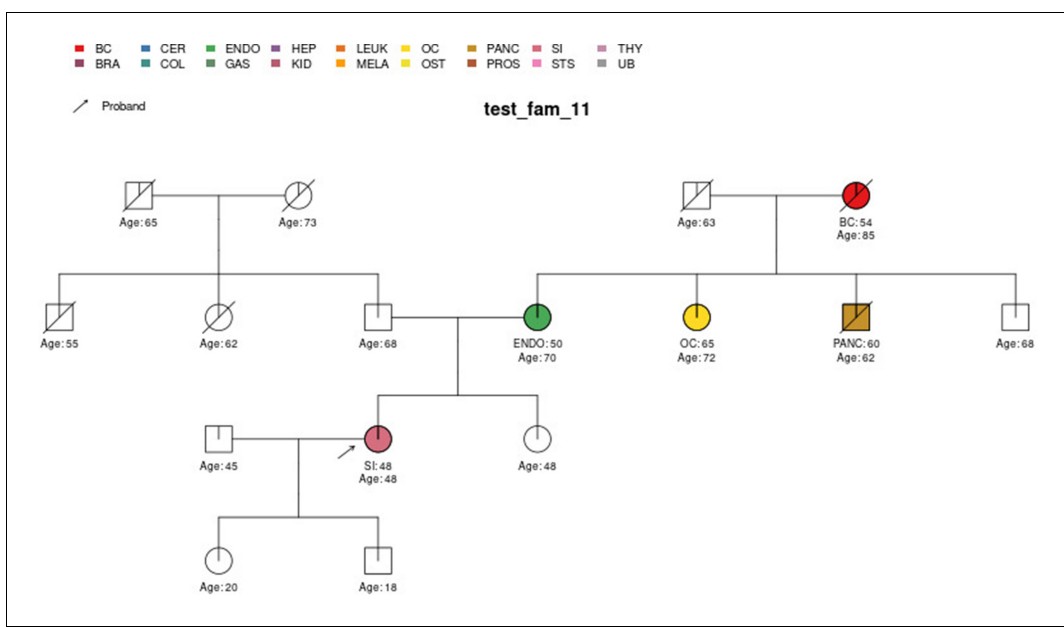

**Figure 9.** Sample pedigree test_fam_11.
The online version of this article includes the following figure supplement(s) for figure 9:

**Figure supplement 1.** PanelPRO output with test_fam_11 as pedigree input.
**Figure supplement 2.** BRCAPRO output with test_fam_11 as pedigree input.
**Figure supplement 3.** MMRPRO output with test_fam_11 as pedigree input.
**Figure supplement 4.** IBIS output with test_fam_11 as pedigree input.
**Figure supplement 5.** PREMM-5 output with test_fam_11 as pedigree input.

**Table 3.** List of main functions in PanelPRO.

| Category | Name | Description |
|---|---|---|
| Pre-processing | checkFam | Checks family structure as defined by the user. The inputs are a `data.frame` specifying the pedigree and a built database returned by buildDatabase. The output is a modified `data.frame` pedigree and list of imputed ages, if missing ages were imputed (see the Missing Data section). |
| Pre-processing | buildDatabase | Subsets the internal database PanelPRODatabase depending on the cancers and genes selected. The input is the list PanelPRODatabase. The output is another list which is a subset of PanelPRODatabase. |
| Algorithm | PanelPROCalc | Estimates the posterior carrier probabilities and future risks of the proband. The inputs are the outputs of checkFam. The outputs are lists of posterior probabilities and future risks for the proband. |
| Main function | PanelPRO | Runs main function. The inputs are the user-specified pedigree, a vector of cancers in the model, a vector of genes in the model, and other optional parameters. The output is a list of estimates of posterior carrier probabilities for each genotype, along with future cancer risks and ranges for each of these. |

Two pedigrees that illustrate the differences between PanelPRO and PREMM-5 are *test_fam_7* (*Figure 5*) and *test_fam_11* (*Figure 9*). For *test_fam_7*, PanelPRO estimates a 42.3% probability of carrying an MLH1, MSH2, MSH6, PMS2, or EPCAM mutation (when excluding the possibility of multiple simultaneous gene mutations), compared to 32% for PREMM-5. This pedigree only contains family history of colorectal and endometrial cancers, which PREMM-5 uses as key risk factors, leading to similar results. In contrast, the mutation probability estimates between the same two models for *test_fam_11* are quite different. This pedigree is an extreme example that contains history of endometrial, small intestine, ovarian, and pancreatic cancers, but PREMM-5 groups the latter three cancers into a single risk factor for any other Lynch syndrome-associated cancers. The differences in the model approaches and assumptions result in PanelPRO giving a 93% estimate for a mutation in any of the aforementioned genes (without multiple simultaneous gene mutations), while PREMM-5 returns 3.2%. *test_fam_11* is an extreme pedigree, but it nonetheless illustrates the flexibility of PanelPRO for incorporating very detailed pedigree information with a high clinical impact.

## Implementation summary

We list the key functions with their input(s) and output(s) in *Table 3*. The PanelPRO function calls the pre-processing functions and the algorithm engine in the back-end, so we expect that most users will only need to use this main function. However, the other functions can be called separately if desired. For example, users can call buildDatabase to inspect the database of model parameters or run checkFam to examine the pedigree after it has been checked.

## Methods: Mendelian modeling

In this section, we give the mathematical details of the main PanelPROCalc engine, which encompasses approximating genotype distributions of counselees and their future cancer risks.

## Genotype probabilities

PanelPRO predicts an individual's probability of having a specified genotype. We use the notation in *Table 5*. Without loss of generality, let the subscript $i = 1$ represent the counselee (i.e. the individual who is counseled). For simplicity, we only consider one counselee, although the model can handle multiple counselees in a computationally efficient manner. The counselee's genotype probability is:

$$P(\mathbf{G}_1 \,|\, \mathbf{H}, \mathbf{U}). \tag{1}$$

Using Bayes' rule, the law of total probability and the assumption of independence of family phenotypes given genotypes and sex, this can be written as

$$
\begin{aligned}
P(\mathbf{G}_1 \,|\, \mathbf{H}, \mathbf{U}) \; &\propto P(\mathbf{G}_1) \sum_{\mathbf{G}_2, \ldots, \mathbf{G}_I} \prod_{i=1}^{I} P(\mathbf{H}_i \,|\, \mathbf{G}_i, U_i) P(\mathbf{G}_2, \ldots, \mathbf{G}_I \,|\, \mathbf{G}_1) \\
&= P(\mathbf{G}_1) \sum_{\mathbf{G}_2, \ldots, \mathbf{G}_I} \prod_{r=1}^{R} \prod_{i=1}^{I} P(\mathbf{H}_{ri} \,|\, \mathbf{G}_i, U_i) P(\mathbf{G}_2, \ldots, \mathbf{G}_I \,|\, \mathbf{G}_1).
\end{aligned}
\tag{2}
$$

From this representation of the posterior probability, we can clearly see the model and user inputs to PanelPRO. $P(\mathbf{G}_1)$ represents the allele frequencies for each gene in the model. $P(\mathbf{H}_{ri} \,|\, \mathbf{G}_i, U_i)$ are derived from the cancer penetrances $P(T_{ri} = t \,|\, \mathbf{G}_i, U_i)$. Explicitly,

**Table 5.** Notation for Mendelian Modeling for a model with $K$ genes and $R$ cancers and a family of $I$ members. The subscript $i$ denotes the $i$ th family member.

| Variable and notation | Description | R object from user input, if applicable |
|---|---|---|
| Genotypes | | |
| $\mathbf{G}_i = (G_{ki})_{k=1}^{K}$ | Genotype of individual $i$, where $G_{ki}$ is the binary indicator for carrying a deleterious mutation in the $k$ th gene | |
| $\mathbf{G} = (\mathbf{G}_i)_{i=1}^{I}$ | Genotypes of all family members $i = 1, \ldots, I$ | |
| Sex | | |
| $U_i$ | Binary indicator that individual $i$ is male | Sex |
| $\mathbf{U} = (U_i)_{i=1}^{I}$ | Binary male indicators for all family members $i = 1, \ldots, I$ | |
| Cancer history | | |
| $T_{ri}$ | Age of diagnosis of the $r$ th cancer for individual $i$ | AgeXX |
| $C_i$ | Individual $i$'s censoring age (current age or age of death) | CurAge |
| $\delta_{ri} = \mathbb{I}(T_{ri} \leq C_i)$ | Binary indicator that cancer $r$ occurs before the censoring age for individual $i$ | |
| $\mathbf{H}_{ri} = \begin{cases} (C_i, \delta_{ri}) & \text{if } \delta ri=0 \\ (C_i, \delta_{ri}, T_{ri}) & \text{if } \delta ri=1 \end{cases}$ | Observed history of the $r$ th cancer for individual $i$, not including risk modifiers and interventions | |
| $\mathbf{H}_i = (\mathbf{H}_{ri})_{r=1}^{R}$ | All observed history for individual $i$ | |
| $\mathbf{H} = (\mathbf{H}_i)_{i=1}^{I}$ | Observed histories for all family members $i = 1, \ldots, I$ | |
| $T_{d,ri}$ | Individual $i$'s age of death from causes other than cancer $r$ | |
| $T_{ri}^* = \min(T_{ri}, T_{d,ri})$ | Individual $i$'s age of first outcome, either cancer $r$ or death from causes other than cancer $r$ | |
| $J_{ri} = I(T_{ri}^* = T_{ri})$ | Binary indicator that individual $i$ develops the $r$ th cancer | isAffXX |

$$P(\mathbf{H}_{ri} \mid \mathbf{G}_i, U_i) = \begin{cases} 1 - \sum_{s=1}^{C_i} P(T_{ri} = s \mid \mathbf{G}_i, U_i) & \text{if } \delta ri=0 \\ P(T_{ri} = T_{ri}^{obs} \mid \mathbf{G}_i, U_i) & \text{if } \delta ri=1 \end{cases}$$

where $T_{ri}$ is the random variable and $T_{ri}^{obs}$ is the observed cancer age. By default, the allele frequencies and penetrances are obtained from existing peer-reviewed studies and estimates, but are completely customizable within PanelPRO.

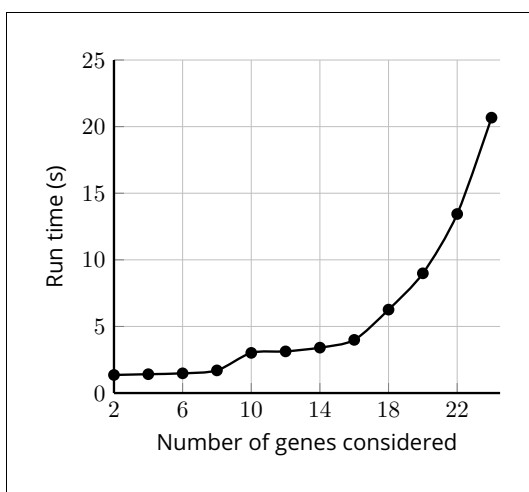

**Figure 10.** Sample run-times for *test_fam_1* evaluated by PanelPRO on the default settings, as a function of the number of genes considered. The paring parameter is set to 2. These run time experiments were performed on a 2020 Linux machine with an 11th Gen Intel(R) i7-1165G7 chip at 2.80 GHz.

Since the genotype space $\{(\mathbf{G}_2, \ldots, \mathbf{G}_I) : \mathbf{G}_i \in \{0,1\}^K, i = 2, \ldots, I\}$ is large for large values of $K$, we use the peeling-paring algorithm (*Madsen et al., 2018*) as an approximation, only allowing a pre-specified number of mutations to be simultaneously present in the same individual. The pedigree structure from the user input is used to derive the $P(\mathbf{G}_2, \ldots, \mathbf{G}_I \mid \mathbf{G}_1)$ term in *Equation* 2 using Mendelian laws of inheritance.

### Future cancer risk

PanelPRO also estimates future cancer risk, based on the previously calculated genotype distribution of the individual. Suppose the counselee has not developed the $r$ th cancer by their current age. Then the risk of developing the $r$ th cancer in $t_0$ years is

$$P\big(T_{r1}^* \le C_1 + t_0, J_{r1} = 1 \mid \mathbf{H}, \mathbf{U}\big) = \sum_{\mathbf{G}_1} P\big(T_{r1}^* \le C_1 + t_0, J_{r1} = 1 \mid \mathbf{G}_1, \mathbf{U}\big) P(\mathbf{G}_1 \mid \mathbf{H}, \mathbf{U}). \tag{3}$$

*Equation 3* produces so-called 'crude' risk, since competing risks of death from causes other than the specified cancer are accounted for. Thus, the reported future risk is the probability that the counselee develops the $r$ th cancer within the next $t_0$ years and does not die from other causes beforehand, given the cancer history and sexes of the family. $P\big(T_{r1}^* \le C_1 + t_0, J_{r1} = 1 \mid \mathbf{G}_1, \mathbf{U}\big)$ is the crude penetrance and is also a model input with default values estimated from the literature.

PanelPRO also provides the option to report 'net' future risk, which is the probability that the counselee develops the $r$ th cancer in a hypothetical world where they cannot die from other causes, given the cancer history and sexes of the family. This risk type is not as realistic but some clinicians find it useful, as it focuses on the specified cancer and allows them to factor qualitatively the patient-specific covariates that may affect the patient's risk. To report net future risk, PanelPRO uses the net penetrances $P(T_{ri} = t \mid \mathbf{G}_i, U_i)$. Note that the genotype probabilities in Equation 2 were calculated using net penetrances, as we do not collect death from other causes as a user input.

## Discussion

PanelPRO is a highly flexible package which provides an interface to efficiently calculate carrier probabilities for a wide array of cancer susceptibility genes, as well as future cancer risks. It is designed for R users. Similarly to the BayesMendel package, it can provide the computational engine behind clinical and counseling decision support tools.

It excels in being fully customizable. Any combination of the 24 genes and 18 cancers currently in version 0.2.0 of the package can be included in the model. New genes and cancers can easily be added, and in fact the code allows for an arbitrary number of genes and cancers. Risk modifiers have been included for certain procedures, and more can be added as additional information becomes available. The user can also change the internal database of parameter values.

The package includes a comprehensive check on the input pedigree to ensure users are informed of potentially inconsistent or infeasible data entries. When it is possible to do so safely, the data is automatically remedied and the user is then notified. Otherwise, the program will halt with an informative error message. Once the pedigree is pre-processed, the posterior probabilities are calculated efficiently. For example, *test_fam_1*, which has 19 members and family history of 2 cancers, runs with all the default settings in a few seconds as shown in *Figure 10*. The polynomial run-time of the peeling-paring algorithm is alleviated with PanelPRO's Rcpp implementation. Even when relaxing the maximum mutations (paring) parameter, the C++ implementation is able to handle the calculations efficiently. Run-times in these ranges are certainly appropriate for clinical use, as well as use in a research setting where possibly hundreds of pedigrees have to be processed through PanelPRO. Moreover, the peeling-paring algorithm run-time scales linearly in the number of family members in the pedigree and can handle hundreds of members in an inter-generational configuration easily.

PanelPRO has two main limitations. Firstly, the initial release does not handle pedigrees which contain loops. This additional functionality would be desirable in future releases, although loops in pedigrees do not happen frequently. Several studies suggest either exact or approximate computations for pedigrees with loops, see *Stricker et al., 1995* and *Totir et al., 2009*. Secondly, the polynomial scaling of peeling-paring as a function of the number of genes considered becomes significant when many genes are incorporated. This issue is of concern because we strive for future

releases to contain far more genes than 24 as data becomes available. Alternative algorithms which have different time complexity properties, such as the Lander-Green family of algorithms (*Lander and Green, 1987*), should be explored. These algorithms scale linearly in terms of the number of genes considered, but are exponential in the number of family members in the pedigree (*Gao et al., 2009*). A future objective for this package is to contain a choice of the carrier probability calculation method, and ideally an automatic selection of the one which is most efficient, depending on family size and total number of genes. Appropriate thresholds of these two parameters need to be determined by a comprehensive benchmarking exercise.

## Acknowledgements

We gratefully acknowledge support from the National Cancer Institute at the National Institutes of Health grants 5T32CA009337 (JWL and TH), 2T32CA009001 (TH), and 4P30CA006516 (GP).

## Additional information

### Funding

| Funder | Grant reference number | Author |
| --- | --- | --- |
| National Institutes of Health | 5T32CA009337 | Jane W Liang<br>Theodore Huang |
| National Institutes of Health | 2T32CA009001 | Theodore Huang |
| National Institutes of Health | 4P30CA006516 | Giovanni Parmigiani |

The funders had no role in study design, data collection and interpretation, or the decision to submit the work for publication.

### Author contributions

Gavin Lee, Software, Investigation, Methodology, Writing - original draft; Jane W Liang, Software, Formal analysis, Validation, Methodology, Writing - review and editing; Qing Zhang, Software, Validation, Investigation, Methodology; Theodore Huang, Conceptualization, Data curation, Software, Investigation, Methodology, Writing - review and editing; Christine Choirat, Supervision, Investigation, Methodology, Project administration, Writing - review and editing; Giovanni Parmigiani, Conceptualization, Formal analysis, Supervision, Methodology, Project administration, Writing - review and editing; Danielle Braun, Conceptualization, Supervision, Methodology, Project administration, Writing - review and editing

### Author ORCIDs

Gavin Lee  https://orcid.org/0000-0003-2659-1163
Jane W Liang  https://orcid.org/0000-0002-2302-3809
Danielle Braun  https://orcid.org/0000-0002-5177-8598

### Decision letter and Author response

Decision letter https://doi.org/10.7554/eLife.68699.sa1
Author response https://doi.org/10.7554/eLife.68699.sa2

## Additional files

### Supplementary files

• Transparent reporting form

### Data availability

This manuscript introduces PanelPRO, an innovative multi-gene multi-cancer Mendelian model. Software for this model, including the model parameter database, is available to download for research use; https://projects.iq.harvard.edu/bayesmendel/panelpro.

The following dataset was generated:

| Author(s) | Year | Dataset title | Dataset URL | Database and Identifier |
|---|---|---|---|---|
| Lee G, Liang JW, Zhang Q, Huang T, Choirat C, Parmigiani G, Braun D | 2021 | PanelPRO R Package | https://projects.iq.har-vard.edu/bayesmendel/panelpro | PanelPRO R Package, projects.iq.harvard.edu/bayesmendel/panelpro |

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

## Appendix 1

### In depth package workflow
Cancer name mapping

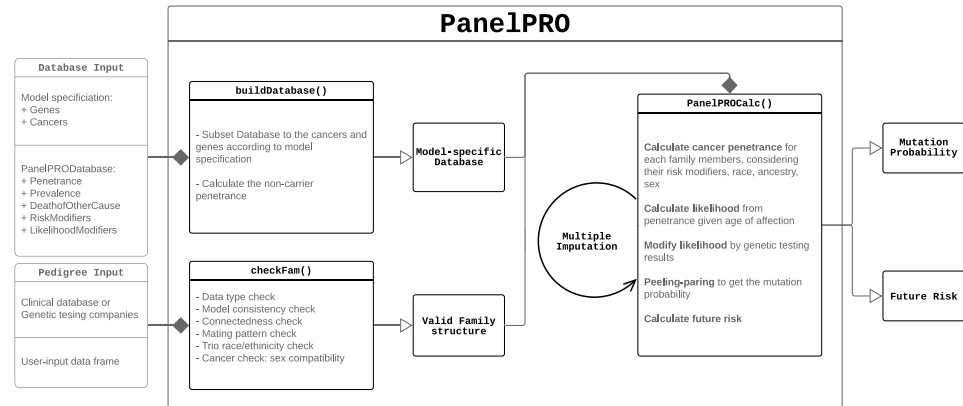

**Appendix 1—figure 1.** PanelPRO in depth package workflow.

**Appendix 1—table 1.** Abbreviations of cancers in PanelPRO.

| Short name | Long name | Short name | Long name |
|---|---|---|---|
| BRA | Brain | OC | Ovarian |
| BC | Breast | OST | Osteosarcoma |
| CER | Cervical | PANC | Pancreas |
| COL | Colorectal | PROS | Prostate |
| ENDO | Endometrial | SI | Small Intestine |
| GAS | Gastric | STS | Soft Tissue Sarcoma |
| KID | Kidney | THY | Thyroid |
| LEUK | Leukemia | UB | Urinary Bladder |
| MELA | Melanoma | HEP | Heptobiliary |

## Sample pedigrees

**Appendix 1—table 2.** Summary of sample pedigrees provided within the package.

| Pedigree name | Number of family members | Cancers present |
|---|---|---|
| test_fam_1 | 19 | BC, OC |
| test_fam_2 | 25 | ENDO, PANC, SI |
| test_fam_3 | 50 | BRA, BC, COL, ENDO, GAS, KID, MELA, OC, PANC, PROS, SI |
| test_fam_4 | 9 | BC, OC, BRA, COL, PROS, ENDO, SI, |
| test_fam_5 | 17 | BC, MELA |
| test_fam_6 | 19 | BC, ENDO, MELA, PANC |
| test_fam_7 | 19 | COL, ENDO |
| test_fam_8 | 20 | COL, PROS |
| test_fam_9 | 19 | BC, PROS |

*Continued on next page*

*Appendix 1—table 2 continued*

| Pedigree name | Number of family members | Cancers present |
| --- | --- | --- |
| test_fam_10 | 21 | BC, COL |
| test_fam_11 | 16 | BC, ENDO, OC, PANC, SI |
| test_fam_12 | 21 | all cancers in *Appendix 1—table 1* |
| err_fam_1 | 10 | BC, PANC |

