## [Decision Letter]

**Acceptance summary:**

The authors have done an admirable job responding to the reviewers and editorial board members. They have increased the number of cases/pedigrees analyzed using this novel modeling tool. They have also outlined how to practically use this program for users.

**Decision letter after peer review:**

Thank you for submitting your article "PanelPRO: An R package for multi-syndrome, multi-gene risk modeling for individuals with a family history of cancer" for consideration by *eLife*. Your article has been reviewed by 2 peer reviewers, one of whom is a member of our Board of Reviewing Editors, and the evaluation has been overseen by a Senior Editor. The reviewers have opted to remain anonymous.

The reviewers have discussed their reviews with one another, and the Editors have drafted this to help you prepare a revised submission.

Essential Revisions:

Overall the reviewers felt that this is outstanding manuscript with potential broad impact in the cancer genetics and oncology community as highlighted in the reviews below. There are a couple of key revisions that are recommended prior to the manuscript being acceptable for publication.

1) Additional examples of cases and the output from the analysis performed on these example cases with comparison when appropriate to existing programs.

2) More details about the practical uses of the program in the clinic and discussion about the ability to import data from existing clinical platforms into PanelPRO.*Reviewer #1:*

This is an outstanding paper that develops and validates a new germline analysis tool for inputting family history and germline sequencing data into a user friendly interface that can be then used to calculate the risk architecture of a given family based upon both ascertained family history information as well as DNA sequencing results.

There are a number of strengths in this manuscript including:

1. The work and program generated here is novel and has a number of advantages to the existing software / platforms currently available to the cancer genetics community – it is more comprehensive in genes that can be inputted and in the family history architecture that can be ascertained.

2. The paper is well written and is able to clearly followed with the methods laid out thoroughly and comprehensively.

3. The impact of this work to the cancer genetics and oncology community will be immediate and I can envision and predict widespread adoption of this software platform in the near term.

One weakness to be noted is that more example cases could be included as test and validation examples and to show the results from the analysis being performed. When appropriate and relevant perhaps a comparison of the results to currently available platforms to analyze germline sequencing data would be of value and strengthen the paper.

*Reviewer #2:*

This work describes an R package called PanelPRO that seeks to assist in identifying individuals at increased risk of cancer due to inherited germline mutations in multiple genes. This tool integrates pedigree data and, importantly, has the ability to be updated as new cancer predisposition genes are discovered and peer-reviewed data on cancer risk becomes available. Patient factors such as risk-reducing surgeries and tumor biomarkers can also be incorporated as part of the risk evaluation. This allows for more personalized risk predictions compared to existing risk prediction models.

Strengths:

This R package offers many advantages over existing programs with similar intended uses. The authors highlight this by noting the limited cancer types and genes considered by existing tools and contrasting this to the capabilities of PanelPRO. Specifically, the authors provide clear examples of this software's ability to incorporate family histories with several different cancer types and to provide an output for prediction of finding a germline pathogenic variant in multiple genes as well as individualized future cancer risk. It also allows users to include individual level risk factors and biomarkers, resulting in a more personalized risk assessment when compared to existing tools. There are several other features that can be changed based on user preference, providing users with a high level of customization. The flexibility to make updates to the R package to reflect changes in the field of cancer genetics is another strength of this work. PanelPRO is designed to be compatible with an existing risk-modeling package, BayesMendel.

The authors provide a clear workflow explaining the use of the package and provide the necessary information to access it in its currently available form. The different variables, notations, and customizable features of the program are clearly explained in the manuscript. Many important variables from the family history that cannot be included in risk assessments using currently available tools can be incorporated into PanelPRO. There are several examples showing different features and capabilities of the program that improve on existing tools. Instructions for use with pedigree information appear to be clear for those familiar with R software. Some limitations of the technology are noted but suggestions to address the described limitations are included.

Weaknesses:

While the strengths of the PanelPRO package are evident, discussion around its use in practice is lacking. By excluding this aspect, potential limitations related to implementation and use are not addressed here, but are key in determining the potential impact of this software in the field. It is not noted if this new package is intended for use in clinical practice, research, or both. Proposed users include users of the existing BayesMendel package, so providing information about users of the BayesMendel package (clinicians versus researchers, volumes, etc) could help readers determine possible applicability of PanelPRO to their own practice. While the authors posit new users may be interested in this software due to its enhanced abilities, further information on new potential users is not included. Due to the absence of discussion of use of this package in a clinical and/or research setting, the likely uptake and impact of this work on the field is difficult to determine. Broader audiences may benefit from more context surrounding existing cancer risk models and their use in cancer genetics to better appreciate the improvements noted here compared to traditional risk modeling programs.

One of the strengths of PanelPRO is the capability for software updates as new knowledge about hereditary cancer syndromes and associated pathogenic variants becomes available. Although this software has the ability to customize the allele frequency and penetrance for a requested gene, meaning it can accommodate predictions for genes not built into its software, built-in gene data relies on published, peer-reviewed data for allele frequencies and penetrance. Additional context surrounding how new genes get added to PanelPRO would help readers understand the significance of the work.

While the improvements compared to existing programs are clear, context around the current use of available risk models in practice, and specific examples of intended use, would help the reader better appreciate the potential significant impact of PanelPRO in clinical and/or research cancer genetics settings. Information about the ability to import data directly from popular pedigree programs would also help determine the potential uptake and impact.

---

## [Author Response]

Essential Revisions:Overall the reviewers felt that this is outstanding manuscript with potential broad impact in the cancer genetics and oncology community as highlighted in the reviews below. There are a couple of key revisions that are recommended prior to the manuscript being acceptable for publication.1) Additional examples of cases and the output from the analysis performed on these example cases with comparison when appropriate to existing programs.

We appreciate the reviewer's thoughtful comments. We agree that additional example cases and comparisons would strengthen the manuscript. We have added eight additional example pedigrees to PanelPRO version 0.2.0, with a total of thirteen pedigrees now provided as examples in the package (see revised manuscript lines 103-108 in the User Input subsection). These pedigrees cover a broad range of realistic and extreme examples that illustrate the utility of the PanelPRO model. In the manuscript, we have added a new subsection to the Methods section titled Additional Examples (see revised manuscript lines 236-273). We visualize five of these sample pedigrees, as well as their outputs based on PanelPRO with all the default settings (Figures 5-9 and their corresponding figure supplements). Where appropriate, depending on the cancers reported in the example pedigree, we compare these outputs to the outputs of BRCAPRO (breast and ovarian) and MMRPRO (colorectal and endometrial), both from the BayesMendel R package (Chen et al., 2004); CanRisk (breast and ovarian), also known as BOADICEA (Lee et al., 2019; Carver et al., 2021); IBIS (breast and ovarian) (Tyrer et al., 2004); and PREMM-5 (colorectal, endometrial, and any other Lynch syndrome-associated cancers) (Kastrinos et al., 2017). We have added Table 3 to the main text which summarizes the cancer inputs and output for each of these models. The advantage of PanelPRO is that it takes as input multiple cancers; the output includes both carrier probability and future risk estimates; future risks of cancers are shown over time; and the entire output figure is interactive, meaning that numerical estimates can be sought by hovering over the relevant data point(s).

2) More details about the practical uses of the program in the clinic and discussion about the ability to import data from existing clinical platforms into PanelPRO.

PanelPRO is currently available for research purposes (https://projects.iq.harvard.edu/bayesmendel/panelpro) and we may license it in the future for commercial clinical use. The existing BayesMendel package is also available for research purposes (https://projects.iq.harvard.edu/bayesmendel/bayesmendel-r-package) and is currently licensed for commercial clinical use to CRA Health, CancerGene Connect, Progeny, FamHis, MagView, Igentify, CancerIQ, and Finch genetics. For BayesMendel, we leave clinical integration to the software licensees, including the integration into electronic health records using software such as EPIC. We envision a similar dissemination plan for PanelPRO. Since R is capable of reading in a wide range of data types, from flat files to compressed data objects, there are many approaches for importing and formatting PanelPRO-compatible pedigrees from existing clinical platforms. We do hope, funding permitted, to develop a R shiny app to support a user interface for R usage of the package, but we leave integration of the PanelPRO model into existing clinical platforms to the companies who specialize in this. We have added a discussion on the practical usage of PanelPRO to the end of the Introduction section in lines 78-85.

Reviewer #1:One weakness to be noted is that more example cases could be included as test and validation examples and to show the results from the analysis being performed. When appropriate and relevant perhaps a comparison of the results to currently available platforms to analyze germline sequencing data would be of value and strengthen the paper.

To address this weakness, we have added eight additional pedigrees, for a total of thirteen example cases in PanelPRO version 0.2.0 (lines 103-108 in the User Input subsection) (see also the response to reviewing editor, comment 1). These pedigrees cover a broad range of realistic and extreme examples that illustrate the utility of the PanelPRO model. In the manuscript, we provide visualizations for five of these sample pedigrees, as well as the results from applying PanelPRO with all the default settings to these pedigrees (Figures 5-9 and their corresponding figure supplements). We compare the PanelPRO results for these pedigrees to those of BRCAPRO and MMRPRO (from the BayesMendel R package) (Chen et al., 2004), CanRisk (also known as BOADICEA) (Lee et al., 2019; Carver et al., 2021), IBIS (Tyrer et al., 2004) and PREMM-5 (Kastrinos et al., 2017), when relevant (i.e. for pedigrees with cancer family history supported by both models). The output from PanelPRO provides both carrier probability and future risk estimates; future risks of cancers are shown over time; and the entire figure is interactive, meaning that numerical estimates can be sought by hovering over the relevant data point(s). Lines 236-273 and Table 3 in the newly-added “Additional Examples" section summarize this analysis and discussion.

A pair of such examples illustrates the differences between PanelPRO and PREMM-5 for estimating the risk for gene mutations associated with Lynch syndrome. For the pedigree `test_fam_7`, the mutation probabilities for any of the genes MLH1, MSH2, MSH6, PMS2 or EPCAM is 42.3% according to PanelPRO (when excluding the possibility of multiple simultaneous gene mutations), whereas it is 32% for PREMM-5. This pedigree only contains history of colorectal and endometrial cancers, which PREMM-5 uses as key risk factors in their model, leading to similar results (though PanelPRO gives estimates of each individual gene in addition). In contrast, the mutation probability estimates between the same two models for `test_fam_11` are quite different. This pedigree is an extreme example for illustration purposes and is unlikely to appear in clinical settings. Nevertheless, it contains endometrial, small intestine, ovarian and pancreatic cancers, all of which are associated with Lynch syndrome, but the latter three are grouped, along with other cancers, into a single risk factor in the PREMM-5 model. It also contains a case of breast cancer in a second-degree relative, but for Lynch syndrome this is not a significant risk factor. The differences in the model approaches, along with differences in assumptions, result in PanelPRO giving a 93% estimate for a mutation in any of the aforementioned genes (without multiple simultaneous gene mutations), whilst PREMM-5 gives 3.2%. This is clearly an extreme pedigree in terms of the number and nature of the relevant cancers, and both models give a recommendation for further genetic evaluation at a cut-o_ of 2.5%; however, it illustrates the exibility of PanelPRO for incorporating very detailed pedigree information that will have a high clinical impact.

Reviewer #2:Weaknesses:While the strengths of the PanelPRO package are evident, discussion around its use in practice is lacking. By excluding this aspect, potential limitations related to implementation and use are not addressed here, but are key in determining the potential impact of this software in the field. It is not noted if this new package is intended for use in clinical practice, research, or both. Proposed users include users of the existing BayesMendel package, so providing information about users of the BayesMendel package (clinicians versus researchers, volumes, etc) could help readers determine possible applicability of PanelPRO to their own practice. While the authors posit new users may be interested in this software due to its enhanced abilities, further information on new potential users is not included. Due to the absence of discussion of use of this package in a clinical and/or research setting, the likely uptake and impact of this work on the field is difficult to determine. Broader audiences may benefit from more context surrounding existing cancer risk models and their use in cancer genetics to better appreciate the improvements noted here compared to traditional risk modeling programs.

The reviewer raises excellent points about the practical usage of PanelPRO (see also the response to reviewing editor, comment 2). The existing BayesMendel package is used in research, as well as integrated into clinical tools such as CancerGene (Chen et al., 2004) (over 4,000 users in more than 75 countries), CRA Health (over 15,000 users per month), Progeny, FamHis, MagView, Igentify, CancerIQ, and Finch genetics. Similarly, we hope PanelPRO will be used by both clinicians and researchers. Family history has long been understood to be a key component for identifying risk and preventing heritable diseases, and clinical tools such as the ones which license BayesMendel are becoming more readily available (Welch et al., 2018). Cancer risk models like PanelPRO and the existing BayesMendel models incorporate family history and other information to quantify risk in terms of carrier probabilities and the future probability of disease development. Beyond current users of the BayesMendel package, we hope that PanelPRO's generality and relevance to panel testing will attract a broader audience in the current landscape for cancer clinical risk assessment. This additional context has been added to the end of the Introduction section in lines 74-75 and 81-85.

One of the strengths of PanelPRO is the capability for software updates as new knowledge about hereditary cancer syndromes and associated pathogenic variants becomes available. Although this software has the ability to customize the allele frequency and penetrance for a requested gene, meaning it can accommodate predictions for genes not built into its software, built-in gene data relies on published, peer-reviewed data for allele frequencies and penetrance. Additional context surrounding how new genes get added to PanelPRO would help readers understand the significance of the work.

As the reviewer notes, one must obtain estimates of the pathogenic variant allele frequency and the penetrance for at least one cancer in the database in order to add a new gene. Therefore, new genes will be added to PanelPRO based on regular literature reviews as conducted in the All Syndromes Known to Man Evaluator (ASK2ME) clinical tool (Braun et al., 2018). The ASK2ME approach identifies best-available studies that adjust for ascertainment; since many papers report odds ratios or relative risks, it then calculates absolute age-specific cancer penetrances when necessary.

In the current PanelPRO database, we incorporate cancer penetrances from data included in the Bayes-Mendel package (primarily from meta-analyses) when available: the BRCA1 and BRCA2 estimates for the probability of developing breast or ovarian cancer (Chen et al., 2020); the MLH1, MSH2, and MSH6 estimates for the probability of developing colorectal or endometrial cancer (Wang et al., 2020; Felton et al., 2007); and the CDKN2A estimates for the probability of developing melanoma (Wang et al., 2010; Begg et al., 2005; Bishop et al., 2002). All other cancer penetrances were pulled from ASK2ME (Braun et al., 2018).

For allele frequencies, we also perform a literature review. In the current PanelPRO database, we used the non-Ashkenazi, Ashkenazi Jewish, and Italian BRCA1 and BRCA2 allele frequency estimates from BRCAPRO (Chen et al., 2004; Antoniou et al., 2002); for MLH1, MSH2, and MSH6, we used the allele frequency estimates from MMRpro (Chen et al., 2004, 2006); and for CDKN2A, we used the allele frequency estimate from Melapro (Chen et al., 2004; Berwick et al., 2006). Allele frequency estimates for ATM, CHEK2, and PALB2 were taken from Lee et al. (2016). The allele frequencies of the remaining genes were estimated based on a 25-gene panel study of 252,223 individuals (Rosenthal et al., 2017) that did not adjust for ascertainment. In this case, we rescaled the reported estimates by the ratio of the ascertained and unascertained allele frequencies for a gene reported in both our existing database and the study. We now include notes on adding new genes and obtaining model parameter estimates, including new citations, in lines 131-150 of the Model Input subsection.

While the improvements compared to existing programs are clear, context around the current use of available risk models in practice, and specific examples of intended use, would help the reader better appreciate the potential significant impact of PanelPRO in clinical and/or research cancer genetics settings. Information about the ability to import data directly from popular pedigree programs would also help determine the potential uptake and impact.

We thank the reviewer for their recommendations (see also the response to reviewing editor, comment 2 and the response to reviewer 2, comment 1). As mentioned in previous responses, for BayesMendel, we leave clinical integration to the software licensees, including the integration of electronic medical records such as EPIC. We envision a similar dissemination plan for PanelPRO. Since R is capable of reading in a wide range of data types, from flat files to compressed data objects, there are many approaches for directly importing pedigrees from existing clinical platforms. We do hope, funding permitted, to develop an R shiny app to support R usage of the package that requires less technical skill, but we leave integration of the PanelPRO model into existing clinical platforms to the companies who specialize in this. We have added a discussion on the practical usage of PanelPRO to the end of the Introduction section in lines 78-85.

References

Antoniou, A., Pharoah, P., McMullan, G., Day, N., Stratton, M., Peto, J., Ponder, B., and Easton, D. (2002). A comprehensive model for familial breast cancer incorporating BRCA1, BRCA2 and other genes. British journal of cancer, 86(1):76.

Begg, C. B., Orlow, I., Hummer, A. J., Armstrong, B. K., Kricker, A., Marrett, L. D., Millikan, R. C., Gruber, S. B., Anton-Culver, H., Zanetti, R., et al. (2005). Lifetime risk of melanoma in CDKN2A mutation carriers in a population-based sample. Journal of the National Cancer Institute, 97(20):1507{1515.

Berwick, M., Orlow, I., Hummer, A. J., Armstrong, B. K., Kricker, A., Marrett, L. D., Millikan, R. C., Gruber, S. B., Anton-Culver, H., Zanetti, R., et al. (2006). The prevalence of CDKN2A germ-line mutations and relative risk for cutaneous malignant melanoma: an international population-based study. Cancer Epidemiology and Prevention Biomarkers, 15(8):1520{1525.

Bishop, D. T., Demenais, F., Goldstein, A. M., Bergman, W., Bishop, J. N., Paillerets, B. B.-d., Chompret, A., Ghiorzo, P., Gruis, N., Hansson, J., et al. (2002). Geographical variation in the penetrance of CDKN2A mutations for melanoma. Journal of the National Cancer Institute, 94(12):894{903.

Braun, D., Yang, J., Gri_n, M., Parmigiani, G., and Hughes, K. S. (2018). A clinical decision support tool to predict cancer risk for commonly tested cancer-related germline mutations. Journal of genetic counseling,27(5):1187{1199.

Carver, T., Hartley, S., Lee, A., Cunningham, A. P., Archer, S., de Villiers, C. B., Roberts, J., Ruston, R., Walter, F. M., Tischkowitz, M., et al. (2021). Canrisk tool|a web interface for the prediction of breast and ovarian cancer risk and the likelihood of carrying genetic pathogenic variants. Cancer Epidemiology and Prevention Biomarkers, 30(3):469{473.

Chen, J., Bae, E., Zhang, L., Hughes, K., Parmigiani, G., Braun, D., and Rebbeck, T. R. (2020). Penetrance of breast and ovarian cancer in women who carry a BRCA1/2 mutation and do not use risk-reducing salpingo-oophorectomy: An updated meta-analysis. JNCI cancer spectrum, 4(4):pkaa029.

Chen, S., Wang, W., Broman, K. W., Katki, H. A., and Parmigiani, G. (2004). BayesMendel: an R environment for mendelian risk prediction. Statistical applications in genetics and molecular biology, 3(1):1{19.

Chen, S., Wang, W., Lee, S., Nafa, K., Lee, J., Romans, K., Watson, P., Gruber, S. B., Euhus, D., Kinzler, K. W., et al. (2006). Prediction of germline mutations and cancer risk in the Lynch syndrome. Jama, 296(12):1479{1487.

Felton, K., Gilchrist, D., and Andrew, S. (2007). Constitutive deficiency in DNA mismatch repair: is it time for Lynch III? Clinical genetics, 71(6):499{500.

Kastrinos, F., Uno, H., Ukaegbu, C., Alvero, C., McFarland, A., Yurgelun, M. B., Kulke, M. H., Schrag, D., Meyerhardt, J. A., Fuchs, C. S., et al. (2017). Development and validation of the premm5 model for comprehensive risk assessment of lynch syndrome. Journal of clinical oncology, 35(19):2165.

Lee, A., Mavaddat, N., Wilcox, A. N., Cunningham, A. P., Carver, T., Hartley, S., de Villiers, C. B., Izquierdo, A., Simard, J., Schmidt, M. K., et al. (2019). Boadicea: a comprehensive breast cancer risk prediction model incorporating genetic and nongenetic risk factors. Genetics in Medicine, 21(8):1708{1718.

Lee, A. J., Cunningham, A. P., Tischkowitz, M., Simard, J., Pharoah, P. D., Easton, D. F., and Antoniou, A. C. (2016). Incorporating truncating variants in PALB2, CHEK2, and ATM into the BOADICEA breast cancer risk model. Genetics in Medicine, 18(12):1190.

Rosenthal, E. T., Bernhisel, R., Brown, K., Kidd, J., and Manley, S. (2017). Clinical testing with a panel of 25 genes associated with increased cancer risk results in a significant increase in clinically significant findings across a broad range of cancer histories. Cancer genetics, 218:58{68.

Tyrer, J., Du_y, S. W., and Cuzick, J. (2004). A breast cancer prediction model incorporating familial and personal risk factors. Statistics in medicine, 23(7):1111{1130.

Wang, C., Wang, Y., Hughes, K. S., Parmigiani, G., and Braun, D. (2020). Penetrance of colorectal cancer among mismatch repair gene mutation carriers: A meta-analysis. JNCI Cancer Spectrum.

Wang, W., Niendorf, K. B., Patel, D., Blackford, A., Marroni, F., Sober, A. J., Parmigiani, G., and Tsao, H. (2010). Estimating CDKN2A carrier probability and personalizing cancer risk assessments in hereditary melanoma using MelaPRO. Cancer research, 70(2):552{559.

Welch, B. M., Wiley, K., Pfieger, L., Achiangia, R., Baker, K., Hughes-Halbert, C., Morrison, H., Schiffman, J., and Doerr, M. (2018). Review and comparison of electronic patient-facing family health history tools. Journal of genetic counseling, 27(2):381{391.